# Mismanagement of SARS-CoV-2 Infection Pre Hospitalisation during the Omicron Era: Antibiotics and Steroids Instead of Early Antivirals

**DOI:** 10.3390/v16071005

**Published:** 2024-06-22

**Authors:** Andrea Giacomelli, Cosmin Lucian Ciubotariu, Martina Zacheo, Andrea Rabbione, Margherita Pieruzzi, Federico Barone, Andrea Poloni, Giacomo Casalini, Giacomo Pozza, Marta Colaneri, Matteo Passerini, Anna Lisa Ridolfo, Cristina Gervasoni, Dario Cattaneo, Andrea Gori, Spinello Antinori

**Affiliations:** 1Department of Biomedical and Clinical Sciences (DIBIC), Università degli Studi di Milano, Via G.B. Grassi 74, 20157 Milan, Italy; andrea.giacomelli@unimi.it (A.G.); cosmin.ciubotariu@unimi.it (C.L.C.); martina.zacheo@unimi.it (M.Z.); andrea.rabbione@unimi.it (A.R.); margherita.pieruzzi@unimi.it (M.P.); federico.barone@unimi.it (F.B.); andrea.poloni@unimi.it (A.P.); giacomo.pozza@unimi.it (G.P.); cristina.gervasoni@asst-fbf-sacco.it (C.G.); andrea.gori@unimi.it (A.G.); 2III Division of Infectious Diseases, ASST Fatebenefratelli Sacco, Luigi Sacco Hospital, 20157 Milan, Italy; giacomo.casalini@asst-fbf-sacco.it (G.C.); annalisa.ridolfo@asst-fbf-sacco.it (A.L.R.); 3II Infectious Diseases Unit, Ospedale Luigi Sacco, ASST Fatebenefratelli Sacco, 20157 Milan, Italy; marta.colaneri@gmail.com (M.C.); matteo.passerini@unimi.it (M.P.); 4Department of Pathophysiology and Transplantation, Università degli Studi di Milano, 20157 Milan, Italy; 5Gestione Ambulatoriale Politerapie Outpatient Clinic, ASST Fatebenefratelli Sacco University Hospital, 20154 Milan, Italy; dario.cattaneo@asst-fbf-sacco.it; 6Department of Infectious Diseases, ASST Fatebenefratelli Sacco University Hospital, 20154 Milan, Italy; 7Centre for Multidisciplinary Research in Health Science (MACH), Università degli Studi di Milano, 20157 Milan, Italy

**Keywords:** Omicron, nirmatrelvir/ritonavir, inappropriate treatment, COVID-19, antibiotics, steroids

## Abstract

The aim of this study was to assess the prevalence of inappropriate treatment among hospitalised patients affected by SARS-CoV-2 infection before hospital admission during the Omicron era. This single-centre, retrospective observational study included all the patients hospitalised because of SARS-CoV-2 infection during three periods characterised by the Italian prevalence of an Omicron variant of concern: (1) January–May 2022 (BA.1–BA.2), (2) June–October 2022 (BA.5), and (3) November 2022–March 2023 (BQ.1-XBB). Inappropriate treatment was defined as pre-hospitalisation exposure to antibiotics and/or steroids in the absence of a documented bacterial infection or the need for steroid treatment of an underlying medical condition. A total of 931 subjects were hospitalised: 394 in period 1, 334 in period 2, and 203 in period 3. Of the 157 patients undergoing inappropriate treatment (16.9%), 142 (15.3%) received antibiotics and 52 (5.6%) steroids. The proportion of inappropriately treated patients significantly decreased over time, from 23.1% in period 1 to 11.7% in period 2 and 13.3% in period 3 (*p* < 0.001), and there was a parallel decrease in antibiotic (*p* < 0.001) and steroid treatment (*p* < 0.013). Only 13 subjects (1.4%) received early pre-hospitalisation treatment for SARS-CoV-2. A significant proportion of hospitalised COVID-19 patients were exposed to inappropriate treatment before hospital admission.

## 1. Introduction

The evolution of the SARS-CoV-2 pandemic led the World Health Organisation to declare the end of the COVID-19 global health emergency in May 2023 [1]. The key factors shaping this evolution included the widespread introduction of mass vaccinations against SARS-CoV-2 [2,3,4] and the emergence of more transmissible but less pathogenic variants of concern (VoCs) [5], the convergence of which ensured that most people had some form of immunity against SARS-CoV-2 [6]. Consequently, in this Omicron era, the subjects at risk of developing severe disease requiring hospitalisation were predominantly the elderly [7], the unvaccinated [8], and people with immunological disorders [9,10]. Furthermore, the early out-of-hospital use of monoclonal antibodies (sotrovimab) and antivirals (remdesivir, nirmatrelvir/ritonavir) also reduced the risk of disease progression [11], although this required the early confirmation of SARS-CoV-2 infection by means of a rapid antigenic or molecular test and prompt access to outpatient care. Randomised clinical trials have consistently demonstrated that this approach reduces hospitalisation and/or death rates [12,13,14], but it may be hindered by logistical constraints such as the parenteral routes for monoclonal antibodies and remdesivir and concerns about potential drug interactions, such as boosting for nirmatrelvir/ritonavir, which may limit their accessibility [15,16,17]. 

The drawback of the potentially suboptimal use of early antiviral treatments against SARS-CoV-2 has, since the early phase of the pandemic, been compounded by a disturbing trend towards the over-use of antibiotic [18] and steroid treatments [19] despite the lack of scientific evidence supporting their efficacy [20,21]. This trend is still relatively widespread in clinical practice despite the current availability of early treatments for at-risk COVID-19 patients. The aim of this study was to assess the prevalence of the pre-hospital use of inappropriate medications among patients hospitalised because of SARS-CoV-2 infection during the Omicron era. 

## 2. Materials and Methods

### 2.1. Study Design and Setting

This single-centre, retrospective observational study included all the patients hospitalised in the Infectious Diseases Department and Intensive Care Unit (ICU) of Milan’s Luigi Sacco Hospital because of SARS-CoV-2 infection between 1 January 2022 and 30 March 2023, enrolled in the Luigi Sacco COVID-19 registry [7,22,23,24,25,26].

### 2.2. Study Population

The study population consisted of all the patients aged ≥ 18 years with a positive nasopharyngeal antigen or molecular test for SARS-CoV-2, except for those who were diagnosed during hospitalisation or came from residential healthcare facilities or other medical institutions.

### 2.3. Data Collection

As previously described in detail [7,22,23,24,25,26], the epidemiological, clinical, radiological, and laboratory data of patients hospitalised with SARS-CoV-2 infection were prospectively recorded in an electronic database. For the purposes of this study, data concerning their pre-hospital exposure to antibiotics and steroids from the time of symptom onset and their concomitant medications at the time of hospitalisation were retrospectively retrieved from their patient records.

### 2.4. Aims

The primary aim of this study was to estimate the proportion of COVID-19 patients receiving inappropriate antibiotic and/or steroid treatment before hospital admission. The secondary aims were to assess the impact of inappropriate treatment on the oxygen requirements at the time of hospital admission and estimate the proportion of missed opportunities for early antiviral treatment with nirmatrelvir/ritonavir before hospital admission. 

### 2.5. Definitions

This study considered three epidemic periods based on the prevalent Omicron VoC circulating in Italy at the time: (1) from 1 January to 30 May 2022 (BA.1–BA.2); (2) from 1 June 2022 to 30 October 2022 (BA.5); and (3) from 1 November 2022 to 30 March 2023 (BQ.1-XBB) [27].

The patients were divided into two groups on the basis of whether they were diagnosed as having SARS-CoV-2 (by means of an antigen or molecular nasopharyngeal swab test) before or at the time of hospitalisation.

Disease severity at the time of admission was defined on the basis of the five categories specified in the most recent NIH guidelines [28]: (1) no COVID-19 symptoms; (2) mild.: the presence of any of the various signs and symptoms of COVID-19 (e.g., fever, cough, sore throat, malaise, headache, muscle pain, nausea, vomiting, diarrhoea, loss of taste and/or smell in the absence of shortness of breath, dyspnoea, or abnormal chest imaging; (3) moderate: clinical or imaging evidence of lower respiratory tract disease and oxygen saturation (SpO_2_) of ≥94% as measured in room air at sea level by means of pulse oximetry; (4) severe: SpO_2_ of <94% in room air at sea level, an arterial partial pressure of oxygen/fraction of inspired oxygen ratio (PaO_2_/FiO_2_) of <300 mm Hg, a respiratory rate of >30 breaths/min, or a lung infiltration rate of >50%; or (5) critical respiratory failure, septic shock, and/or multiple-organ dysfunction [28].

Inappropriate treatment was defined as pre-hospitalisation exposure to antibiotics and/or steroids in the absence of a documented bacterial infection or the need for steroid treatment of an underlying medical condition [29].

Potential missed opportunities for early antiviral treatment with nirmatrelvir/ritonavir before hospital admission were based on the Italian Medicines Agency (AIFA) indications for early access to nirmatrelvir/ritonavir on the basis of a combination of criteria including age and co-morbidities [29]. Missed opportunities for access to early nirmatrelvir/ritonavir in potentially eligible patients with symptom onsets after the first week of February 2022 (when the drug became available in Italy) [30] were defined as being outside the prescription window (>5 days after symptom onset) or progression to COVID-19 requiring oxygen at the time of hospital admission or increased oxygen support in patients already requiring chronic oxygen support for an underlying medical condition. The patients with a missed opportunity for early access to nirmatrelvir/ritonavir were further stratified on the basis of whether or not they had been diagnosed before hospital admission. 

The presence of co-medications with potential drug–drug interactions or an absolute contraindication to nirmatrelvir/ritonavir use was identified using the Liverpool COVID-19 interaction database [31]. Depending on severity and clinical relevance, an absolute contraindication was assigned when the output of the Liverpool COVID-19 database was a red flag (drug combinations which should be avoided). 

Renal impairment was calculated using the CKD-epi formula, and the patients were stratified on the basis of whether they showed moderate renal impairment (creatinine clearance 30–60 mL/min), severe renal impairment (creatinine clearance < 30 mL/min), or a need for chronic renal replacement therapy for those on dialysis.

### 2.6. Statistical Analysis

Continuous variables were recorded as median values with 25th and 75th percentiles, and categorical variables as frequencies and percentages.

The proportion of patients receiving inappropriate treatment was estimated overall, by stratification on the basis of the period of hospital admission, and by stratification on the basis of whether they had been diagnosed before or at the time of hospital admission. 

The proportion of missed opportunities for early nirmatrelvir/ritonavir treatment was evaluated overall and by stratification on the basis of the period of hospital admission.

Multivariable logistic regression analyses were used to assess the impact of inappropriate treatment on disease severity at hospital admission (defined as COVID-19 with any form of oxygen requirement) while accounting for age, biological sex, and the period of hospital admission. Associations were estimated using adjusted odds ratios (aORs) with their 95% confidence intervals (CIs). Statistical significance was set at 0.05. The analyses were made using the R-Studio software (v. 4.2.3).

### 2.7. Ethics Statement

This study was approved by our local Ethics Committee [Comitato Etico Interaziendale Area 1, Milan, Italy (Protocol No. 16088)].

## 3. Results

### 3.1. Characteristics of the Study Population

Of the 931 subjects hospitalised because of SARS-CoV-2 infection, 394 were hospitalised in period 1 (BA.1–BA.2), 334 in period 2 (BA.5), and 203 in period 3 (BQ.1-XBB) (Table 1). 

During this study, the proportion of patients aged over 65 years increased from 75.4% in period 1 to 82.6% in period 2 and 90.1% in period 3 (*p* < 0.001), and the same was true of the proportion of subjects who had received ≥2 doses of vaccine: from 66.6% to 85.1% and 86.7% (*p* < 0.001). The median time from symptom onset to hospital admission significantly declined from six days (IQR 2–10) in period 1 to three days (IQR 2–6) in period 2 and three days (IQR 2–7) in period 3 (*p* < 0.001), as did the proportion of subjects with critical disease (from 15.2% in period 1 to 5.1% in period 2 and 6.4% in period 3: *p* < 0.001) and the proportion of subjects diagnosed before hospital admission (from 48.2% in period 1 to 38% in period 2 and 36.9% in period 3: *p* < 0.001). In-hospital mortality remained significant throughout the study period (17.3% in period 1, 11.1% in period 2, 17.2% in period 3: *p* = 0.040). COVID-19 patients diagnosed before hospital admission showed a longer time from symptoms onset to hospitalisation when compared to those diagnosed in the Emergency Department [6 (IQR 3–19) vs. 3 (IQR 1–6); *p* < 0.001].

### 3.2. Inappropriate Treatment before Hospitalisation

Overall, 157 of our COVID-19 patients (16.9%) received inappropriate pre-hospitalisation treatment: 142 (15.3%) were exposed to antibiotics and 52 (5.6%) to steroids. The proportion of inappropriately treated patients significantly decreased over time from 23.1% in period 1 to 11.7% in period 2 and 13.3% in period 3 (*p* < 0.001), and there was a parallel decrease in both antibiotic (*p* < 0.001) and steroid treatment (*p* < 0.013). Table 2 shows the types of inappropriate antibiotic treatment: macrolides were the most frequently used (67/931, 7.2%), followed by amoxicillin +/− clavulanic acid (38/931, 4%) and fluoroquinolones (25/931, 2.7%). COVID-19 patients diagnosed before hospital admission received a higher proportion of improper antibiotic and steroid treatments compared to those diagnosed at the time of hospital admission [105/392 (26.8%) vs. 37/539 (6.9%), *p* < 0.001, and 42/392 (10.7%) vs. 10/539 (1.9%), *p* < 0.001, respectively].

### 3.3. Logistic Regression Analysis of the Factors Associated with Disease Severity at the Time of Hospital Admission

Multivariable logistic regression analysis showed that age (aOR 1.02 [95%CI 1.01–1.03] per each additional year) was independently associated with higher odds of being hospitalised with more severe disease, whereas being hospitalised during period 2 was associated with lower odds than being hospitalised in period 1 (aOR 0.39 [95%CI 0.25–0.59]) (Table 3).

### 3.4. Missed Opportunities for Early Antiviral Treatment with Nirmatrelvir/Ritonavir

Only 13/931 subjects (1.4%) received appropriate early pre-hospitalisation treatment for SARS-CoV-2, the frequency of which slightly increased over time: antivirals from 0.3% in period 1 to 0.9% in period 2 and 3% in period 3 (*p* = 0.009), and monoclonal antibodies from 0.3% in period 1 to 0% in period 2 and 1.5% in period 3 (*p* = 0.031).

The reasons for the missed opportunities for pre-hospitalisation access to nirmatrelvir/ritonavir are shown in Figure 1.

After excluding the subjects who had been hospitalised before nirmatrelvir/ritonavir was available, 189 of the 729 patients hospitalised with SARS-CoV-2 (25.9%) were red flagged for an absolute contraindication by the Liverpool COVID-19 interaction database, 100 (13.7%) were asymptomatic upon admission, 10 (1.4%) received early pre-hospitalisation therapy, and 3 (0.4%) did not have any risk factor for disease progression (indicating eligibility for nirmatrelvir/ritonavir according to the AIFA guidelines). Of the remaining 427 potentially eligible patients, 270 (63.2%) missed the opportunity for pre-hospitalisation treatment because they were not within the treatment window of five days from symptom onset (139/427, 32.5%) or because their SARS-CoV-2 infection progressed to requiring oxygen ≤5 days after symptom onset (131/427, 30.7%). Among the 270 who missed the opportunity for pre-hospitalisation treatment, 157 (58.1%) were not tested for SARS-CoV-2 before hospital admission: 63 (45.3%) of the 139 admitted >5 days after symptom onset, and 94 (71.8%) of the 131 who required oxygen ≤5 days after symptoms onset. The high proportion of subjects not tested before hospital admission did not significantly vary throughout this study, with it being 62.2% in period 1, 54.9% in period 2, and 59.5% in period 3 (*p* = 0.587).

## 4. Discussion

Our study shows that a significant proportion of the patients hospitalised because of SARS-CoV-2 infection during the Omicron era were exposed to inappropriate antibiotic or steroid treatment before hospital admission and simultaneously missed opportunities for receiving early antiviral treatment.

The characteristics of our hospitalised patients were similar to those of the general population of elderly subjects with co-morbidities who are more prone to developing a severe form of SARS-CoV-2 infection and its related complications [7], just as immunocompromised subjects are particularly susceptible to developing severe COVID-19 [9,10]. On the other hand, the decreasing severity of the disease upon hospital admission observed in our cohort appears to stem from an evolution of the epidemiological picture, which is characterised by the increasing population coverage of the primary vaccine cycle and subsequent booster doses [8], and the hybrid immunity gained as a result of previous natural infections [6]. More specifically, during the period of the circulation of the BA.1 and BA.2 VoCs, Italy experienced daily case numbers of up to 180,000, which led to a more significant burden on the healthcare system than that observed in subsequent periods [32], and longer intervals between symptom onset and hospitalisation. However, despite the trend towards reduced disease severity upon hospital admission over time, the continuing significant impact of SARS-CoV-2 infection on frail subjects is highlighted by the unchanging 15% overall case fatality rate, which is particularly concerning as our findings also show that fewer than 2% of the patients in our study cohort received appropriate early treatment before hospital admission. This is unfortunately in line with the data of the national registry of early COVID-19 treatments showing the under-use of all early treatments in Italy [33]. 

There are a number of possible reasons for this failure. First of all, although it has been shown that early antiviral treatments (particularly nirmatrelvir/ritonavir treatment) are also effective in previously vaccinated subjects at risk of disease progression [11,34], the randomised controlled trials (RCTs) available during the early phase of their introduction only provided information concerning their efficacy in unvaccinated subjects [13], and this may have led patients and physicians to misperceive the risk of disease progression in vaccinated subjects. Secondly, difficulties in administering some drugs requiring a parenteral route (i.e., remdesivir and monoclonal antibodies) may have reduced their uptake [17]. Thirdly, fear of potential drug–drug interactions with a ritonavir booster [16,31] combined with the lack of self-confidence of general practitioners in prescribing antivirals [15,16] and the bureaucracy involved in prescribing them [27,32,33] may have limited their use. Finally, access to early antiviral treatment relies on a prompt diagnosis of SARS-CoV-2 outside the hospital setting, whereas we found that two-thirds of our patients who missed the opportunity for early treatment with nirmatrelvir/ritonavir before hospital admission had not undergone early testing. This was an unexpected finding given the widespread availability of rapid antigen testing during the study period, but it may have been at least partially due to a lack of awareness of the disease and a mistaken underestimation of its risk over time [35,36,37].

Another unexpected finding after almost two years of the COVID-19 pandemic was the high prevalence of inappropriate pre-hospitalisation antibiotic and steroid treatments. Shockingly, one in five subjects had received treatments that have been shown to be ineffective against SARS-CoV-2 infection, and, although the rate of inappropriate treatment decreased over time, it was still more than 10% during the last epidemic period characterised by the circulation of the BQ.1-XBB VoC.

The inappropriateness of using antibiotics in subjects with SARS-CoV-2 infection was first documented soon after the start of the pandemic in hospitalised and non-hospitalised subjects [18]. Previously, the outpatient visits of 30% of Medicare beneficiaries with COVID-19 had been associated with antibiotic prescriptions, 50.7% of which were for azithromycin [38]. This strategy was partially justified by the lack of data regarding the prevalence of bacterial co-infections, but observational studies have since shown that this prevalence does not support the empirical use of antibiotics in subjects with COVID-19 [39]. The indiscriminate use of macrolides (also the most frequently prescribed type of antibiotic among our patients) seems to have been related to non-experimental studies, suggesting that they may play a role in reducing disease severity [40], but it is now known that azithromycin does not offer any survival benefit to patients with SARS-CoV-2 infection [41,42,43]. 

We found that most antibiotic prescriptions (especially during the period of circulating BA.1 and BA.2 VoCs) were given to subjects who had been diagnosed on the basis of a nasopharyngeal swab before hospital admission. This is in line with AIFA’s national data concerning the use of antibiotics outside hospitals, which showed a worrying increase in the consumption of antibiotics (especially macrolides) during the first semester of 2022 in comparison with 2021 [44]. The same report underlined the fact that this increase was all the more unexpected, as a regulatory note released in April 2020 during the initial wave of the pandemic in Italy had established that the use of these antibiotics for unauthorised purposes could only be considered within the framework of RCTs [45]. Thus, in January 2022, the AIFA further underlined that no antibiotic, i.e., azithromycin, was effective against COVID-19, strongly arguing against its indiscriminate use [46].

The situation seems to be different in the case of the inappropriate use of steroids, which have been shown to be effective in reducing mortality in patients with severe COVID-19 requiring oxygen [47]. However, no beneficial effect has been observed in subjects with less severe disease, and their use during the early phase of a viral infection could expose patients to the untoward effects of steroids and the risk of immunosuppression [19].

We did not find a significant association between inappropriate treatments and disease severity upon hospital admission after adjusting for a set of known confounders, but it can be speculated that the combination of such factors would have a detrimental effect on the quality of patient care. For example, a significant proportion of our patients were exposed to inappropriate treatment for a viral infection instead of receiving the available early SARS-CoV-2 treatment: i.e., up to two out of three patients potentially eligible for early treatment with nirmatrelvir/ritonavir missed an opportunity for early antiviral access before being admitted to the hospital with SARS-CoV-2 infection in all three study periods.

### Study Limitations

The retrospective nature of this study exposed it missing data, so we cannot exclude the possibility that the rate of inappropriate medication might have been underestimated. Secondly, the single-centre design limits the generalisability of our findings to other settings. Thirdly, we were unable to identify the physicians prescribing the inappropriate medications, but, as we excluded subjects diagnosed in other medical facilities or nursing homes, we believe that most of the antibiotic and steroid prescriptions were issued by non-hospital physicians. Fourthly, we could not determine whether the patients who missed an opportunity for early antiviral treatment of SARS-CoV-2 infection had actually declined an offer of treatment before hospital admission. Finally, our study does not provide any insights into the factors driving the inappropriate prescribing. 

## 5. Conclusions

Our study of a frail population of elderly subjects hospitalised because of SARS-CoV-2 infection in the Omicron era showed that most of them missed the opportunity for early access to SARS-CoV-2 treatment and that a significant proportion of them had been exposed to inappropriate antibiotic and/or steroid treatment before hospital admission. There is, therefore, a need to continue informing both the general population and the medical community about the scientific evidence supporting effective means of reducing SARS-CoV-2 morbidity and mortality. 

## Figures and Tables

**Figure 1 viruses-16-01005-f001:**
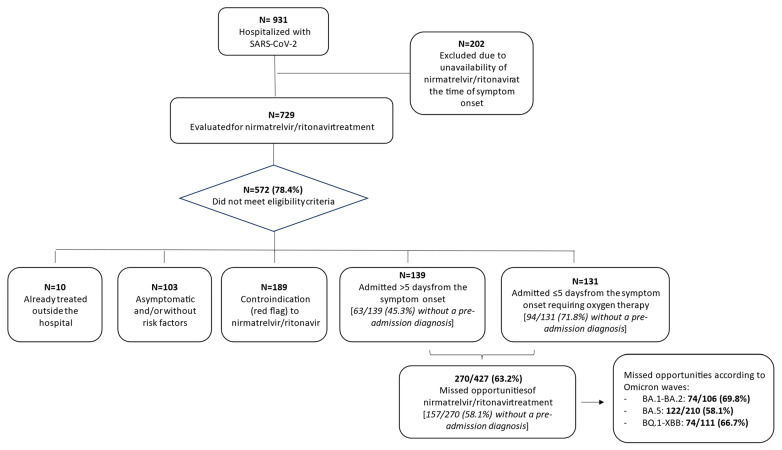
Missed opportunities for accessing nirmatrelvir/ritonavir treatment before hospital admission in patients with SARS-CoV-2 infection.

**Table 1 viruses-16-01005-t001:** Characteristics of the study population according to the different periods of hospital admission. List of abbreviations: n, number; and IQR, Inter Quartile Range.

Characteristics	Total n = 931	BA.1–BA.2 Period n = 394	BA.5 Period n = 334	BQ.1-XBB Period n = 203	*p* Value
**Age, median (IQR)**	81 (70–87)	79 (66–86)	81 (71–88)	83 (75–88)	<0.001
Age > 65, n (%)	756 (81.2)	297 (75.4)	276 (82.6)	183 (90.1)	<0.001
**Male biological sex, n (%)**	500 (53.7)	206 (52.3)	177 (53.0)	117 (57.6)	0.438
**Co-morbidities, n (%)**					
Cardiovascular	297 (31.9)	111 (28.2)	114 (34.1)	72 (35.5)	0.106
Hypertension	575 (61.8)	220 (55.8)	225 (67.4)	130 (64.0)	0.004
Diabetes	208 (22.3)	89 (22.6)	73 (21.9)	46 (22.7)	0.965
Dyslipidaemia	184 (19.8)	64 (16.2)	66 (19.8)	54 (26.6)	0.010
Lung disease	47 (5.0)	33 (8.4)	7 (2.1)	7 (3.4)	<0.001
Cancer	96 (10.3)	36 (9.1)	33 (9.9)	27 (13.3)	0.270
Hematologic neoplasm	67 (7.2)	29 (7.4)	16 (4.8)	22 (10.8)	0.031
Cirrhosis	52 (5.6)	21 (5.3)	19 (5.7)	12 (5.9)	0.952
eGFR mL/min < 30	69 (7.4)	25 (6.3)	22 (6.6)	22 (10.8)	0.107
Dialysis	10 (1.1)	4 (1.0)	1 (0.3)	5 (2.5)	0.061
HIV	9 (1.0)	2 (0.5)	5 (1.5)	2 (1.0)	0.396
Rheumatologic disease	58 (6.2)	29 (7.4)	18 (5.4)	11 (5.4)	0.473
Immunosuppression	95 (10.2)	36 (9.1)	33 (9.9)	26 (12.8)	0.362
Obesity	101 (10.8)	48 (12.2)	37 (11.1)	16 (7.9)	0.273
**Median vaccine doses, n (IQR)**	3 (2–3)	2 (0–3)	3 (3–3)	3 (3–4)	<0.001
**Previous COVID-19, n (%)**	44 (4.7)	13 (3.3)	13 (3.9)	18 (8.9)	0.006
**Days from symptom onset to admission, median (IQR)**	4 [2,3,4,5,6,7,8]	6 [2,3,4,5,6,7,8,9,10]	3 [2,3,4,5,6]	3 [2,3,4,5,6,7]	<0.001
**SARS-CoV-2 diagnosis before hospital admission, n (%)**	392 (42.1)	190 (48.2)	127 (38.0)	75 (36.9)	0.005
**Oxygen requirement at hospital admission, n (%)**	491 (52.7)	263 (66.8)	124 (37.1)	104 (51.2)	<0.001
**Disease severity at admission, n (%)**					<0.001
No COVID-related symptoms	114 (12.2)	56 (14.2)	40 (12.0)	18 (8.9)	
Mild	326 (35.0)	75 (19.0)	170 (50.9)	81 (39.9)	
Moderate	300 (32.2)	150 (38.1)	79 (23.7)	71 (35.0)	
Severe	101 (10.8)	53 (13.5)	28 (8.4)	20 (9.9)	
Critical	90 (9.7)	60 (15.2)	17 (5.1)	13 (6.4)	
**Inappropriate treatment before hospital admission, n (%)**					
Antibiotic treatment	142 (15.3)	84 (21.3)	35 (10.5)	23 (11.3)	<0.001
Steroid treatment	52 (5.6)	32 (8.1)	11 (3.3)	9 (4.4)	0.013
Antibiotic treatment and/or steroid, n (%)	157 (16.9)	91 (23.1)	39 (11.7)	27 (13.3)	<0.001
**Early antiviral before hospital admission, n (%)**	9 (10)	1 (0.3)	3 (0.9)	5 (3.0)	0.009
Nirmatrelvir/ritonavir	4 (0.4)	1 (0.3)	1 (0.3)	2 (1.0)	
Remdesivir	2 (0.2)	0 (0.0)	1 (0.3)	1 (0.5)	
Molnupiravir	3 (0.4)	0 (0.0)	1 (0.3)	2 (1.5)	
**Early monoclonal before hospital admission, n (%)**	4 (0.4)	1 (0.3)	0 (0.0)	3 (1.5)	0.031
Sotrovimab	1 (0.1)	1 (0.3)	0 (0.0)	0 (0.0)	
Tixagevimab/cilgavimab	3 (0.3)	0 (0.0)	0 (0.0)	3 (1.5)	0.016
**Early intra-hospital SARS-CoV-2 treatment, n (%)**	207 (22.2)	40 (10.2)	101 (30.2)	66 (32.5)	<0.001
**Intra-hospital death, n (%)**	140 (15.0)	68 (17.3)	37 (11.1)	35 (17.2)	0.040

**Table 2 viruses-16-01005-t002:** Types of inappropriate antibiotic treatments used before hospital admission in patients with SARS-CoV-2 infection.

	Total n = 931	BA.1-BA.2 Period n = 394	BA.5 Period n = 334	BQ.1-XBB Period n = 203	*p*-Value
**Antibiotic treatment, n (%)**	142 (15.3)	84 (21.3)	35 (10.5)	23 (11.3)	<0.001
Amoxicillin +/− clavulanic acid	38 (4.0)	23 (5.8)	9 (2.7)	6 (2.9)	0.025
Cephalosporins	19 (2.0)	9 (2.3)	5 (1.5)	5 (2.5)	0.673
Macrolides	67 (7.2)	43 (10.9)	15 (4.5)	9 (4.4)	<0.001
Trimethoprim-sulfamethoxazole	1 (0.1)	1 (0.3)	0 (0.0)	0 (0.0)	0.505
Fluoroquinolones	25 (2.7)	14 (3.6)	5 (1.5)	6 (3.0)	0.223
Unspecified antibiotic	6 (0.6)	2 (0.5)	4 (1.2)	0 (0.0)	0.220

**Table 3 viruses-16-01005-t003:** Logistic regression of factors associated with O2 requirement at hospital admission. List of abbreviation: aOR, adjusted odds ratio.

	aOR	95% CI	*p* Value
**Age (per 1 year more)**	1.02	1.01–1.03	0.0005
**Male vs. female**	1.36	0.98–1.90	0.062
**Improper treatment before admission**	1.55	0.98–2.50	0.063
**BA.5 vs. BA.1-BA.2 period**	0.39	0.25–0.59	<0.0001
**XBB-BQ.1 vs. BA.1-BA.2 period**	0.67	0.42–1.05	0.085

## Data Availability

Data will be made available on request.

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
