# Peer review of "Mismanagement of SARS-CoV-2 Infection Pre Hospitalisation during the Omicron Era: Antibiotics and Steroids Instead of Early Antivirals"

_viruses, 2024, doi:10.3390/v16071005_

Round 1

Reviewer 1 Report

Comments and Suggestions for Authors

In the present study, the authors assessed the prevalence of inappropriate treatment among hospitalized patients affected by SARS-CoV-2 infection before hospital admission during the Omicron eraa and performed a thorough analysis of the characteristics of the study population in an Italian hospital. 

Major comments

- It should be mentioned that this is a descriptive study that refers to the infection control policies of a single hospital. 

. Azithromycin has been widely used in the management of COVID-19.(please see at: Ayerbe L, Risco-Risco C, Forgnone I, Pérez-Piñar M, Ayis S. Azithromycin in patients with COVID-19: a systematic review and meta-analysis. J Antimicrob Chemother. 2022 Feb 2;77(2):303-309. doi: 10.1093/jac/dkab404. PMID: 34791330; PMCID: PMC8690025). In Table 2 of the present study macrolides were the most frequently used (67/931, 7.2%) and considered as inappropriate antibiotic treatment. Where there any guidlines and if so, when they were applied for treatment of COVID-19 in Italy? 

- In lines 101-103 the authors state that ''The patients were divided into two groups on the basis of whether they were diagnosed as having SARS-CoV-2 (by means of an antigen or molecular nasopharyngeal swab test)  before or at the time of hospitalisation. Are there any data to investigate the time of testing and the onset of treatment (i.e. is late testing associated with inappropriate tretment?)

Minor comments:

- Figure 1. Please provide a figure of  better quality.

Comments on the Quality of English Language

The manuscript needs minor editing in English.

Author Response

please se attachement

Reviewer 2 Report

Comments and Suggestions for Authors

I congratulate you on completing your etrospective observational study and writing up the results. This study deal with a pretty well known issue in Italy, Mismanagement of SARS-CoV-2 infection in home patients. I have some comments and questions.

Comment 1. I believe the authors missed some opportunities to include a citation the AIFA's recommendations for home management of Covid19: https://www.aifa.gov.it/documents/20142/1616529/EN_Raccomandazioni_AIFA_gestione_domiciliare_COVID-19_Vers10_10.03.2023.pdf

Author Response

please se attachement
